# SAFRAN: An interpretable, rule-based link prediction method outperforming embedding models

**Simon Ott**                                   SIMON.OTT@MEDUNIWIEN.AC.AT
*Institute of Artificial Intelligence and Decision Support, Medical University of Vienna, Austria*

**Christian Meilicke**                          CHRISTIAN@INFORMATIK.UNI-MANNHEIM.DE
*Data and Web Science Research Group, University Mannheim, Germany*

**Matthias Samwald**                            MATTHIAS.SAMWALD@MEDUNIWIEN.AC.AT
*Institute of Artificial Intelligence and Decision Support, Medical University of Vienna, Austria*

## Abstract

Neural embedding-based machine learning models have shown promise for predicting novel links in knowledge graphs. Unfortunately, their practical utility is diminished by their lack of interpretability. Recently, the fully interpretable, rule-based algorithm AnyBURL yielded highly competitive results on many general-purpose link prediction benchmarks. However, current approaches for aggregating predictions made by multiple rules are affected by redundancies. We improve upon AnyBURL by introducing the SAFRAN rule application framework, which uses a novel aggregation approach called Non-redundant Noisy-OR that detects and clusters redundant rules prior to aggregation. SAFRAN yields new state-of-the-art results for fully interpretable link prediction on the established general-purpose benchmarks FB15K-237, WN18RR and YAGO3-10. Furthermore, it exceeds the results of multiple established embedding-based algorithms on FB15K-237 and WN18RR and narrows the gap between rule-based and embedding-based algorithms on YAGO3-10.

## 1. Introduction

Link prediction in knowledge graphs is a vibrant area of research and development and encompasses a wide range of different approaches. Currently, most state-of-the-art approaches use low-dimensional representations (embeddings) of knowledge graphs to predict new knowledge. However, as such approaches are black boxes, the validity of their predictions can be questioned due to lack of interpretability. Recent research has focused on learning symbolic rules from knowledge graphs that try to detect correlations of entities in knowledge bases and express them as first-order Horn rules. In contrast to neural models, such rule-based methods are fully transparent and predictions can be explained intuitively.

However, while current research on rule-based approaches puts a lot of emphasis on designing rule-mining algorithms, research on methods for applying learned rules and aggregating their predictions in a meaningful way is comparatively neglected.

AnyBURL [Meilicke et al., 2019, 2020] is currently one of the best performing symbolic rule learners and demonstrated results competitive with embedding methods on link prediction benchmarks such as FB15k-237. However, the application of rules generated by AnyBURL is currently limited by functional redundancies, making it difficult to meaningfully aggregate predictions made by different rules and thereby limiting predictive performance.

In this paper, we introduce the high-performance rule application framework SAFRAN[1] ('Scalable And fast non-redundant rule application') that employs a novel aggregation approach. We propose an algorithm called Non-redundant Noisy-OR that detects redundant rules prior to aggregation, mitigating their negative effects and improving predictive performance. This algorithm for aggregating predictions is not necessarily restricted to rules learned by AnyBURL and could also be applied to other rule-based systems.

## 2. Preliminaries

### 2.1 Link prediction

Knowledge graphs contain facts, such as "Amsterdam is the capital of the Netherlands" or "Marie Curie was born in Warsaw", which are represented as semantic triples containing a subject (head $h$), predicate (relation $r$) and object (tail $t$). Formally, a knowledge graph is a directed heterogeneous multigraph $G = (E, R, T)$ where E and R are the set of entities and relations respectively and T is a set of triples $\{(h, r, t)\} \subseteq E \times R \times E$. The task of link prediction is to predict either the tail of a given head and relation $(h, r, ?)$ or the head of a given tail and relation $(?, r, t)$ such that the resulting triple describes a correct fact which has not been in $G$ before.

### 2.2 AnyBURL

We now give an overview about the main AnyBURL algorithm, which is explained in detail in [Meilicke et al., 2019] and [Meilicke et al., 2020]. AnyBURL (Anytime Bottom Up Rule Learning) is a novel walk-based method and is currently one of the best symbolic rule learners, competing with current state-of-the-art embedding approaches [Rossi et al., 2021]. Walk-based (bottom-up) methods are based on the idea that sampled paths from a knowledge graph (random walks) are examples of very specific rules, which can be transformed into more general rules. Learned rules can then be applied to the knowledge graph for the prediction of novel links between entities.

| | |
|---|---|
| Cyclic (C) | $h(Y, X) \leftarrow b_1(X, A_2), \ldots, b_n(A_n, Y)$ |
| Acyclic 1 (AC1) | $h(c_0, X) \leftarrow b_1(X, A_2), \ldots, b_n(A_n, c_{n+1})$ |
| Acyclic 2 (AC2) | $h(c_0, X) \leftarrow b_1(X, A_2), \ldots, b_n(A_n, A_{n+1})$ |

Table 1: Different types of rules that can be learned by AnyBURL. Uppercase letters represent variables and lowercase letters represent constants. $h(\ldots)$ is called the head, while $b_1(\ldots), \ldots, b_n(\ldots)$ is called the body of a rule. Note that variables or constants of a body atom can be flipped.

AnyBURL learns rules in the form of first-order logic Horn rules of length $n$. In each iteration of the algorithm for mining rules the algorithm samples a random triple $h(c_0, c_1)$, f.e. $lives(max, uk)$ from the training set. Starting from either the head or the tail of this triple, AnyBURL performs a random walk of length $n$, resulting in a ground

---

1. Code can be found at https://github.com/OpenBioLink/SAFRAN

path in the form of $h(c_0, c_1) \leftarrow b_1(c_1, c_2), \ldots, b_n(c_n, c_{n+1})$, f.e. $speaks(max, english) \leftarrow lives(max, uk), lang(uk, english)$. The sampled ground path is subsequently generalized into three predefined types of rules shown in Table 1. Cyclic rules can be generalized from cyclic paths ($c_0 = c_{n+1}$), while AC2 rules can be generalized from acyclic paths ($c_0 \neq c_{n+1}$). AC1 rules are hybrid, as they can be both generalized from cyclic (if $c_0 = c_{n+1}$) and acyclic (if $c_0 \neq c_{n+1}$) paths. All rules that can be generalized from the previous example ground path can be seen in Table 2.

$$\begin{array}{c}
speaks(Y, X) \leftarrow lives(X, A), lang(A, Y) \\
speaks(english, X) \leftarrow lives(X, A), lang(A, english) \\
speaks(Y, max) \leftarrow lives(max, A), lang(A, Y)
\end{array}$$

Table 2: Rules that can be generalized from the ground path $speaks(max, english) \leftarrow lives(max, uk), lang(uk, english)$

The following applies for inferring a set of head triples $\hat{H}_r$ from the body of a rule $r$:

$$\hat{H}_r = \begin{cases} \{h(e_y, e_x) | e_x, e_y \in E \wedge \exists e_2, \ldots, e_n \; b_1(e_x, e_2), \ldots, b_n(e_n, e_y) \in T\}, & \text{if type} = \text{C} \\ \{h(c_0, e_x) | e_x \in E \wedge \exists e_2, \ldots, e_n \; b_1(e_x, e_2), \ldots, b_n(e_n, c_{n+1}) \in T\}, & \text{if type} = \text{AC1} \\ \{h(c_0, e_x) | e_x \in E \wedge \exists e_2, \ldots, e_{n+1} \; b_1(e_x, e_2), \ldots, b_n(e_n, e_{n+1}) \in T\}, & \text{if type} = \text{AC2} \end{cases}$$

AnyBURL calculates the confidence of a rule as the size of the union of inferred triples by a rule and the triples of the training set divided by the number of inferred triples $conf(r) = |\hat{H}_r \cap T|/|\hat{H}_r|$, where $T$ is the set of triples contained in the training set.

## 2.3 Aggregation

Rule-based link prediction methods, such as AnyBURL, produce many rules with different confidences that are subsequently applied to the knowledge graph in order to generate candidates for a link prediction task. Upon a link prediction task $p(s, ?)$, rule $r$ generates the set of predictions $\{y | p(s, y) \in \hat{H}_r\}$ that could act as substitutions for the question mark. Rules can either predict zero, one or multiple entities. As the same entities can be proposed by multiple rules which may differ in confidence, an aggregation of these confidences is required to assess the final confidence of a prediction. Table 3 shows the confidences and results of five fictional rules.

| Rule | Results | Confidence |
|------|---------|-----------|
| $R_0$ | $\{a, b\}$ | 0.9 |
| $R_1$ | $\{c\}$ | 0.8 |
| $R_2$ | $\{c\}$ | 0.7 |
| $R_3$ | $\{b\}$ | 0.3 |
| $R_5$ | $\{a\}$ | 0.1 |
| $R_5$ | $\emptyset$ | 0.1 |

Table 3: Example of different rules generating either zero, one or multiple predictions.

One aggregation approach called *Maximum* aggregation is using the maximum of these confidences [Meilicke et al., 2019]. The score of an entity

$$score(e) = \ max\{conf(r_1), \ldots, \text{conf}(r_k)\}$$

is the maximum confidence of all rules $r_1, \ldots, r_k$ that predict entity $e$. If the maximum confidences of two or more entities are the same, these entities are further ranked by their second best confidence and so on, until all top-k candidates can be distinguished or all rules are processed. This approach results in the following list of ranked candidates when applied to the solutions of Table 3: $ranking_{max} = \langle (b, 0.9), (a, 0.9), (c, 0.8) \rangle$. As the second best rule that proposes $b$ has a higher confidence than the second best rule which proposes $a$, $b$ is ranked before $a$.

Unfortunately, the Maximum aggregation is constrained to relatively simple predictions: Each prediction is primarily informed by only a single rule with the highest confidence; it is not possible to make predictions based on a weighted combination of different rules. These limitations of maximum aggregation are potentially addressed by noisy-or aggregation, first proposed in this context by [Galárraga et al., 2015]. Instead of taking the maximum of confidences, the probability that at least one of the rules proposed the correct candidate is used for generating a list of ranked candidates. The score of an entity $e$ can thus be calculated as

$$score(e) = 1 - \prod_{i=1}^{k}(1 - conf(r_i))$$

where $r_1, \ldots, r_k$ are rules that predict entity $e$. Applying the Noisy-OR approach to the results of Table 3, the entities are ranked as follows: $ranking_{noisy} = \langle (c, 0.94), (b, 0.93), (a, 0.9) \rangle$. This ranking differs from the maximum approach. As $c$ is proposed by the second and third most confident rules, its aggregated confidence is higher than $a$ or $b$, entities proposed by the most confident rule.

In theory, Noisy-OR allows for more sophisticated confidence aggregation. However, it assumes independence/non-redundancy of all rules. As most rules that can be derived from real-world knowledge bases are in some way redundant with other rules, Noisy-OR was found to perform worse than maximum aggregation in practice.

## 2.4 The detrimental effect of redundancy on Noisy-OR aggregation

Redundancies in rules can lead to overestimation of confidences of predicted entities when aggregating using Noisy-OR. As two redundant rules generate the same predictions for the same reasons, the confidences of entities predicted by both rules are overestimated due to double counting. Consider the rules shown in Table 4 where this problem becomes apparent. If the training set contains the ground path $lives(john, uk), lang(uk, english)$, each rule would predict $speaks(john, english)$ upon the prediction task $speaks(john, ?)$ for the same reasons. Aggregating the confidences of all three rules using Noisy-OR would result in an overestimated confidence of 0.988 for the entity *english*.

While these redundancies are apparent, redundancies between rules are often less obvious in practice. Redundancy between two rules can arise from special relationships between relations, such as relations that are nearly equivalent, symmetric relations or relations entailing other relations. Consider the rules shown in Table 5. The first example shows two

| | |
|---|---|
| $speaks(X,Y) \leftarrow lives(X,A), lang(A,Y)$ | 0.9 |
| $speaks(john,Y) \leftarrow lives(john,A), lang(A,Y)$ | 0.7 |
| $speaks(X,english) \leftarrow lives(X,A), lang(A,english)$ | 0.6 |

Table 4: Examples of obvious redundancies between rules.

rules being redundant based on entailment, as every capital of a country is a city within that country. The second example shows redundancy based on the symmetric relationship *married*. Both rules generate predictions for languages spoken by a person, based on what language its spouse speaks. The third example shows that such relationships are not restricted to single relations, but can arise from a combination of relations, as the combination *parent* → *brother* is a duplicate of the relation *uncle*.

| | |
|---|---|
| $citizen(X,Y) \leftarrow born(X,A), city(A,Y)$ | 0.7 |
| $citizen(X,Y) \leftarrow born(X,A), capital(A,Y)$ | 0.7 |
| $speaks(X,Y) \leftarrow married(X,A), speaks(A,Y)$ | 0.6 |
| $speaks(X,Y) \leftarrow married(A,X), speaks(A,Y)$ | 0.6 |
| $lives(X,Y) \leftarrow parent(X,A), brother(A,B), lives(B,Y)$ | 0.6 |
| $lives(X,Y) \leftarrow uncle(X,A), lives(A,Y)$ | 0.6 |

Table 5: Examples of less apparent redundancies between rules.

It is important to understand that there is usually no explicit information available about the symmetry of *married* nor do we know that *capital* is more specific than *city*. It is not even possible to grasp this information from the dataset itself as it might be incomplete and noisy. This makes it impossible to decide redundancy via a procedure that makes use of relevant background knowledge.

## 3. Approach

### 3.1 Algorithm

To overcome the problem of redundancy when using the Noisy-OR aggregation method, we propose an approach to cluster rules based on their redundancy degree prior to aggregation. Predictions of rules in a cluster are aggregated using the Maximum approach, as this approach is not susceptible to redundancies. Predictions of the different clusters are then further aggregated using the Noisy-OR approach. We call the resulting aggregation technique *Non-redundant Noisy-OR*. A pseudocode of Non-redundant Noisy-OR can be seen in Appendix G.

As a metric for redundancy between two rules $r_i$, $r_j$ the Jaccard Index $sim(r_i, r_j) = |\hat{H}_{r_i} \cap \hat{H}_{r_j}|/|\hat{H}_{r_i} \cup \hat{H}_{r_j}|$ of the sets of inferred triples is used. As the calculation of the Jaccard coefficient is very inefficient for large sets, the Jaccard coefficient is estimated using the MinHash scheme [Broder, 1997], which makes time complexity linear and memory usage constant.

Naturally rules are pre-clustered by their relation of the head atom, as two rules having a different relation in the head are never used in the same prediction task and the union of their inferred triples is always empty $h_{r_i} \neq h_{r_j} \rightarrow \hat{H}_{r_i} \cap \hat{H}_{r_j} = \emptyset$. Thus, similarity matrices of rules are calculated for each head relation and used to cluster the rules.

Two rules $r_i$, $r_j$ are considered *redundant* and assigned to the same cluster if $sim(r_i, r_j) > t$, where $t$ is a threshold. However a global threshold $t$ is not capable of defining the cutoff point for redundancy for all relations, rule types and directions of prediction (prediction of head entities or tail entities). As the sets of rules that predict for different relations may vary in distribution of type, length or specificity, the threshold that generates the optimal clustering for a relation may not generate the optimal clustering for a different relation. Furthermore as relations may vary in multiplicity the direction of the prediction has to taken into account. Therefore, a distinct threshold for the prediction of heads and prediction of tails for each relation is required in order to be able to optimally cluster an entire rule set. Furthermore, within a set of rules that predict for the same relation, optimal thresholds may vary between types of rules. As can be seen by the examples in Table 4, the type combination of C and AC1 often tends to form set-subset relation. The Jaccard Index is very insensitve to such relationships. While e.g. a Jaccard index of 0.25 between a rule of type C and a rule of type AC1 may indicate a high redundancy due to the dominance of set-subset relationships within rules of this type combination, it may be considered as a fairly low evidence level for a redundancy between two rules of type C, where such relationships are not as prominent. Thus a differentiation of rule types is needed when defining the cutoff value for redundancy between two rules of a certain relation. For each combination of rule types an independent threshold parameter is used to determine redundancy. Conclusively, the threshold of similarity for rules that predict entities of direction $d$ (head or tail) of a $h$-triple (we use $h$ to refer to a relation, while $r$ is used to refer to rules) is given by $t_{hd} = [t_{C/C}, t_{C/AC1}, t_{C/AC2}, t_{AC1/AC2}, t_{AC1/AC1}, t_{AC2/AC2}]$. Note that if $t_{hd} = [0, 0, 0, 0, 0, 0]$ only the Maximum aggregation is used and if $t_{hd} = [1, 1, 1, 1, 1, 1]$ only Noisy-OR.

An optimal clustering of rules that predict $h$-triples is given by the type combination thresholds $t_{hd}$ that maximize the fitness of the clusters for predicting new links. This fitness is evaluated on the validation set using the mean-reciprocal rank based on the top-k predicted entities. SAFRAN uses one of two search strategies for finding the optimal thresholds: grid search (parameter sweep) and random search. For grid search the range of possible thresholds $[0.0, 1.0]$ is divided by $n$ equally distant steps and each threshold is subsequently used for clustering. For the grid search, we do not distinguish between rule-type specific parameters but use a single relation-specific fix parameter $t_{hd}$ to limit the search space. Contrary to this, the random search randomly samples the six rule-type-specific thresholds of $t_{hd}$.

## 3.2 An Example and its Explanation

Figure 1 demonstrates the effect of our algorithm. It has two parts that both illustrate the list of predicted entities for the prediction task *genre_artist(?, Bryan Adams)*. The upper half of the figure shows the result of applying the Non-redundant Noisy-OR approach, while the lower part presents the results of the maximum aggregation. In addition to the candidate lists some explanations in terms of the rules or rule clusters that predict these entities are

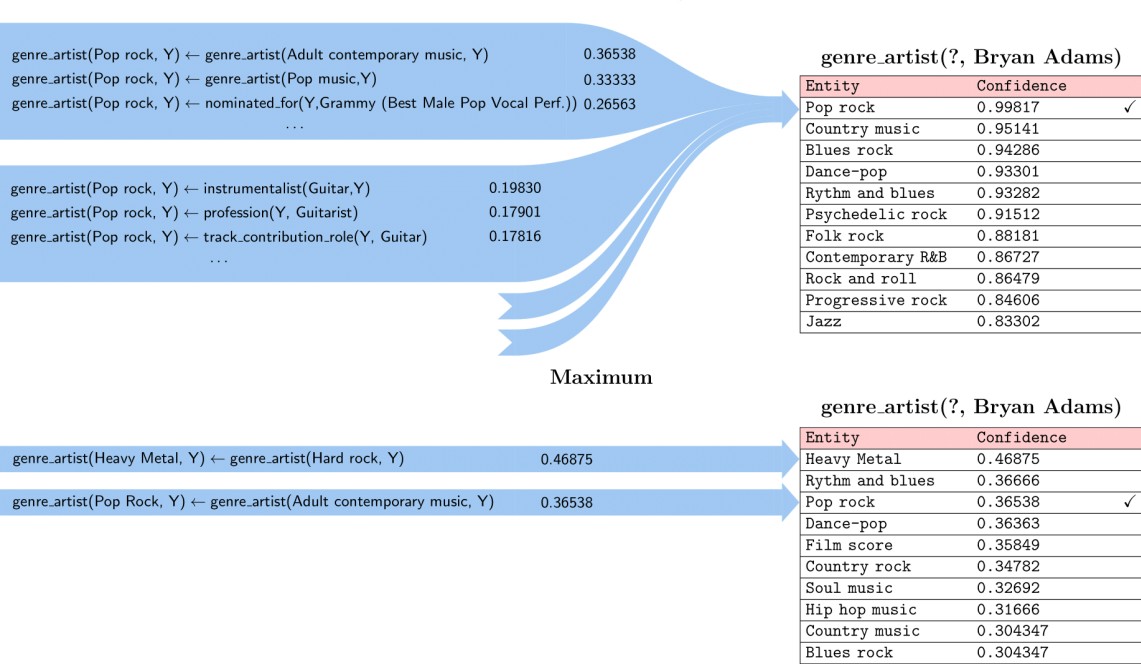

Figure 1: Comparing the list of predicted entities for the prediction task *genre_artist(?, Bryan Adams)* for Non-redundant Noisy-OR and maximum approach.

shown. For the Non-redundant Noisy-OR we depict the two clusters that have the highest maximum confidence. By aggregating these clusters via the Noisy-OR multiplication, the highest rank candidate is *Pop rock*.

A different result can be observed for the maximum aggregation. The correct entity *Pop rock* is ranked at the third position, while the first position is occupied by *Heavy Metal*. The example illustrates nicely that the influence of a single rule can be too strong in the maximum approach, while the Non-redundant Noisy-OR approach allows to aggregate the results in a more meaningful way. This holds in particular if the clusters make sense and combine similar reasons for a prediction within the same cluster. Therefore, in addition to improving predictive performance, our algorithm provides explanations that are easier to understand and that provide deeper insights about the underlying data and the rules derived from them.

## 4. Experiments

We conducted experiments with three established link prediction datasets, including FB15k-237 [Toutanova and Chen, 2015], WN18RR [Dettmers et al., 2018] and YAGO3-10 [Dettmers et al., 2018]. An overview of these datasets can be seen in Table 6. For our experiments we selected only datasets without train-test leakage. We abstained from performing experi-

|  | # Entities | # Relations | # Training | # Test | # Validation |
|---|---|---|---|---|---|
| FB15k-237 | 14,541 | 237 | 272,115 | 17,535 | 20,466 |
| WN18RR | 40,943 | 11 | 86,845 | 3,034 | 3,134 |
| YAGO3-10 | 123,182 | 37 | 1,079,040 | 5,000 | 5,000 |

Table 6: Dataset statistics. Number of entities, relations and triples in each dataset.

ments on the parent datasets FB15k (WN18) [Bordes et al., 2013] of FB15k-237 (WN18RR) as they were found to suffer from heavy testing leakage (f.e. some triples in the test data are present as inverse triples in the training data).

Training, inferencing and evaluations were run on a machine with 24 physical (48 logical) Intel(R) Xeon(R) CPU E5-2650 v4 @ 2.20GHz cores and 264 GB of RAM. As SAFRAN and AnyBURL do not utilize GPU-accelerated computing, only CPUs were used. For optimal comparison with the best results using the maximum aggregation approach presented in [Meilicke et al., 2020], the same learning parameters were adopted for the same datasets. Rulesets used for the results were learned for 1000 seconds each, using 22 threads. For rules learned from WN18RR the maximum length of cyclic rules was set to 5, while for all other datasets a maximum length of 3 was used. The maximum length of acyclic rules was set to 1 for all datasets. As FB15k-237 contains reflexive triples in the form of $r(c, c)$ a flag was set that allows AnyBURL to use such reflexive triples for learning rules. The same rulesets were used for inference and evaluation with AnyBURL and SAFRAN. Random search was performed using a $n$ of 10 and 10000 iterations, while grid search was performed with a $n$ of 200.

We compare our results with the established latent models RESCAL [Nickel et al., 2011], TransE [Bordes et al., 2013], DistMult [Yang et al., 2015], ComplEx [Trouillon et al., 2016], ConvE [Dettmers et al., 2018], RotatE [Sun et al., 2019], TuckER [Balazevic et al., 2019] and the rule-based methods AMIE+ [Galárraga et al., 2015], RuleN [Meilicke et al., 2018], C-NN [Ferré, 2020], Neural LP [Yang et al., 2017], RLvLR [Omran et al., 2018], DRUM [Sadeghian et al., 2019], GPFL [Gu et al., 2020b]. For details on these approaches we refer to their respective original papers. For each approach we report the filtered mean reciprocal rank (MRR), filtered hits@1 and filtered hits@10, where predicted triples that are already known are filtered out, except the test triple itself. The results for FB15k-237, WN18RR and YAGO3-10 can be seen in Table 7. All results reported by us were evaluated according to the average policy [Rossi et al., 2021] (see Appendix A), where the rank of a target entity within a group of same score entities is the average rank of this group. Reported results from [Rossi et al., 2021], [Ruffinelli et al., 2020] were evaluated using the same protocol. Results of C-NN, GPFL, AMIE+, RuleN were evaluated using ordinal (random) policy which results in very similar results than average policy, while to the best of our knowledge there is no information available about the policy used for evaluating RLvLR. Results of DRUM and Neural LP were evaluated using the top policy which is biased towards producing better evaluations results and cannot be compared directly to other results.

Our aggregation method in the more exhaustive random search outperforms both symbolic and subsymbolic approaches on FB15k-237 and WN18RR. This is a surprising result as both data sets have been used for years to evaluate and improve performance of latent methods that are build on the concept of embeddings. On YAGO3-10 the results are

slightly worse. For this data set ComplEx achieves an MRR that is 0.012 higher. However, AnyBURL together with the Non-redundant Noisy-OR is still among the top performing techniques. With our approach we offer a fully interpretable symbolic approach, that allows to explain the results in terms of the rules that generated them, while we are still able to compete with or even outperform latent approaches. The improvement against the AnyBURL default maximum aggregation technique differs a lot between datasets. While we observe a clear improvement of more than 3% in terms of MRR for FB15k-237, the improvement for the other datasets is around 0.5% to 1%. Currently we do not understand the reason for this differences. The most probable explanation is a lower degree of redundancy in the rules that describe the regularities encoded in these datasets.

Another reason for this may be the small validation set sizes of WN18RR and YAGO3-10. While the training set of YAGO3-10 is the biggest of the three datasets used in the experiments, the size of its validation set is only 0.46% of the dataset, compared to 6.6% in FB15k-237. However, non-redundant Noisy-OR needs an appropriate sized validation set that represents a broad spectrum of relations and entities to be able to fully recognize redundancies. The validation set size of WN18RR is 3.4% of the dataset. However as the overall dataset is considerably small, the absolute size of its validation set is only 3,134 triples. As rules generated by AnyBURL can generalize to unseen entities [Ferré, 2020], this size may not be sufficient to generate the most optimal clustering.

| | Approach | | | FB15K-237 | | | WN18RR | | | | YAGO3-10 | | |
| --- | --- | --- | --- | --- | --- | --- | --- | --- | --- | --- | --- | --- | --- |
| | | | MRR | hits@1 | hits@10 | MRR | hits@1 | hits@10 | | MRR | hits@1 | hits@10 |
| Latent | RESCAL | ⋆ | .357 | .263 | .541 | .468 | .439 | .521 | | | | |
| | TransE | ⋆ | .313 | .221 | .497 | .227 | .053 | .526 | ¶ | .501 | .405 | .673 |
| | DistMult | ⋆ | .343 | .250 | .531 | .452 | .413 | .531 | ¶ | .501 | .412 | .661 |
| | ComplEx | ⋆ | .348 | .253 | .534 | .477 | .438 | .543 | ¶ | **.576** | **.505** | **.704** |
| | ConvE | ⋆ | .339 | .248 | .521 | .447 | .411 | .508 | ¶ | .488 | .399 | .657 |
| | RotatE | ¶ | .336 | .238 | .531 | .475 | .426 | .574 | ¶ | .498 | .405 | .670 |
| | TuckER | ¶ | .352 | .259 | .536 | .459 | .430 | .514 | ¶ | .544 | .465 | .680 |
| | HAKE | ▽ | .346 | .250 | **.542** | .497 | .452 | **.582** | | .545 | .462 | .694 |
| Interpretable | C-NN | △ | .296 | .222 | .446 | .469 | .444 | .519 | | | | |
| | DRUM | ‡ | $.343^{\dagger}$ | $.255^{\dagger}$ | $.516^{\dagger}$ | $.486^{\dagger}$ | $.425^{\dagger}$ | $.586^{\dagger}$ | | | | |
| | Neural LP | ◇ | $.240^{\dagger}$ | | $.362^{\dagger}$ | $.435^{\dagger}$ | $.371^{\dagger}$ | $.566^{\dagger}$ | | | | |
| | GPFL | ◇ | .322 | .247 | .504 | .480 | .449 | .552 | | | | |
| | AMIE+ | ♣ | | .174 | .409 | | .358 | .388 | | | | |
| | RuleN | ♣ | | .182 | .420 | | .427 | .536 | | | | |
| | RLvLR | ♯ | .240 | | .393 | | | | | | | |
| AnyBURL with SAFRAN | Maximum | | .355 | .270 | .519 | .494 | .452 | .572 | | .559 | .487 | .686 |
| | Noisy-OR | | .342 | .258 | .502 | .454 | .399 | .562 | | .524 | .444 | .672 |
| | VS | | .363 | .277 | .526 | .497 | .455 | .572 | | .560 | .489 | .689 |
| | NRNO * (grid) | | .370 | .287 | .531 | .501 | .457 | .581 | | .564 | .492 | .691 |
| | NRNO * (random) | | **.389** | **.298** | .537 | **.502** | **.459** | .578 | | .564 | .491 | .693 |

Table 7: MRR, Hits@1, Hits@10 results for FB15K-237, WN18RR and YAGO3-10. Best results for each metric and dataset are marked in bold. *Denotes our approach. †Results were evaluated with top policy for dealing with same score entities (see Appendix A) and are not directly comparable to other approaches. Results marked with ⋆ are from [Ruffinelli et al., 2020], ¶ from [Rossi et al., 2021], △ from [Ferré, 2020], ‡ from [Sadeghian et al., 2019], ◇ from [Gu et al., 2020b], ♣ from [Meilicke et al., 2019], ♯ from [Omran et al., 2018] and ▽ from [Zhang et al., 2020].

In our experiments we used two different search techniques. The random search performs better in most cases. As explained above, it learns thresholds that distinguish between different rule types. Thus, only a random selection of all possible threshold combinations is visited during the search. Nevertheless, type-specific thresholds seem to make sense as the results are in most cases slightly better compared to the grid search, which learns a single threshold per relation.

To support our claim that Non-redundant Noisy-OR could potentially be applied to other rule-based approaches, we performed experiments on all used data sets using rules learned by AMIE+. The results can be seen in Appendix F. We used AMIE in its default setting. This means that it mines only cyclic rules. This could be the reason that we observed no or only a very limited improvement on each data sets with the exception of FB15k-237.

To ascertain the importance of optimal thresholds, we performed another experiment. We evaluated each relation in the validation set using Noisy-OR and Maximum aggregation and apply the approach with the maximum MRR of each relation to the test set. Results for the datasets can be seen in Table 7 denoted as VS. All metrics improved compared to the Maximum aggregation but could not achieve the gain of Non-redundant Noisy-OR.

## 5. Related Work

One of the most prominent rule mining systems is AMIE, which has been described first in [Galárraga et al., 2015] and later, including several extension and improvements in [Lajus et al., 2020]. These papers focus mainly on mining rules (including the confidence computation) rather than their application. This is typical for most rule-mining or inductive logic programming (ILP) papers. The authors touch the topic of how to aggregate confidences only in a single experiment where they compare maximum and the Noisy-OR aggregation. According to their results the Noisy-OR approach works better.

In [Ferré, 2020] another interesting variation of Noisy-OR, the Dempster–Shafer score [Denoeux, 1995], has been applied. This approach is not based on rules, but relies on another symbolic representation using concepts of nearest neighbours. In their experiments the authors found that the aggregation using the Dempster–Shafer score performs at best 1.3 % worse than the maximum score [Ferré, 2020].

It might make sense to decouple the learning of rules from the model used for their application. ProbLogic [De Raedt et al., 2007] offers a well-defined model for applying probabilistic rules to a given set of facts (or triples). ProbLog evaluates probabilistic logic programs by combining *Selective Linear Definite* clause resolution (SLD-resolution) with methods for computing the probability of Boolean formulae. Using ProbLog for applying the rules learned by a rule miner would be similar to the Noisy-OR aggregation we described above. As ProbLog is based on a model-theoretic notion of entailment, predictions created by a rule might enable other rules to fire, which might again result into further predictions (and so on). While this makes sense from a conceptual point of view, such an approach results in runtime problems for larger datasets.

In [Kuželka and Davis, 2020] the authors proposed Markov Logic as a powerful framework for solving knowledge base completion tasks. Instead of learning rules, the authors propose to learn the weights of a Markov logic network. As a consequence, dependencies

between regularities are modeled in a complex way, which would lead to a non-trivial aggregation of evidences. However, the analysis remains entirely theoretical and, so it is doubtful whether the approach is applicable to larger knowledge graphs.

The concept of rule redundancy is crucial for rule aggregation. If two rules predict the same candidate for the same reason, a Noisy-OR aggregation makes no sense. In [Gu et al., 2020a] a rule hierarchy was used to detect redundant rules. However, the application model is based on the maximum aggregation strategy, which means that the removal has no positive impact on the predictive quality. Indeed, the authors focus their experiments on runtime performance and report about a positive impact without a decrease in the predictive quality.

It is important to understand that the notion of rule redundancy is not limited to logical or extentional equivalence. Approaches to detect equivalent duplicate rules are well known in ILP, and have for example been implemented in QuickFOIL [Zeng et al., 2014]. Instead of that, we have to deal with problems where two rules are only partially or nearly redundant.

We note that there are a few approaches that try to generate post-hoc explanations of predictions made by latent models such as knowledge graph embeddings. However, as opposed to AnyBURL/SAFRAN which provides ad-hoc explanations for every predicted triple, such approaches often fail to find explanations. F.e. the approach proposed in [Zhang et al., 2019] applied to the knowledge graph embedding CrossE can only find explanations for 40% of triples in the test set of the FB15K dataset. Furthermore it is not fully transparent if generated explanations actually represent the reason for a prediction of a latent model.

## 6. Conclusion

Interpretable models drastically improve the trust and practical utility of predictions, and help to address current concerns about intransparency and unwanted bias in machine learning models. We introduced a novel method for fully interpretable link prediction that yields new state-of-the-art results among interpretable methods and outperforms state-of-the-art embedding models on some of the established benchmarks. Furthermore, our approach provides explanations that reveal clustering patterns in underlying data and derived rulesets. While our approach can significantly improve predictive performance of rule-based approaches, a more in-depth analysis is needed to assess the quality of generated clusters, which we plan to do in future work. We hope that beyond their direct practical utility, our results help to reinvigorate interest into novel rule-based approaches to complement the embedding- and deep learning-based approaches that have been at the center of attention in machine learning research over the past years.

## Acknowledgments

This work received funding from netidee grant 5171 ('OpenBioLink').

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

## Appendix A. Evaluation metrics and same score policy

We now formally define used metrics in our experiments. Let $t = 2 * |T|$ be the number of prediction tasks in test set $T$ and $rank_i$ be the rank of the correct entity in the predicted list of entities for completion task $i$, then Hits@k and mean-reciprocal rank (MRR) are defined as follows:

$$Hits@k = \frac{1}{t} \sum_{i=1}^{t} 1 \text{ if } rank_i < k$$

$$MRR = \frac{1}{t} \sum_{i=1}^{t} \frac{1}{rank_i}$$

There are multiple policies for dealing with entities that have the same score as the correct entity [Rossi et al., 2021]:

- *Top*: The correct entity is given the best rank among a group of same score entities.

- *Bottom*: The correct entity is given the worst rank among a group of same score entities. This policy is the most conservative.

- *Average*: The rank of the correct entity is the mean rank of all predictions that tie their score with the correct entity.

- *Ordinal*: The rank of the correct entity is given the rank in which it appears in the group of same score entities. This policy usually corresponds to the random policy.

- *Random*: The rank of the correct entity is given a random rank of the ranks within the group of same score entities. This policy usually corresponds to the average policy.

When comparing results from different sources it is important to use comparable policies for dealing with same score entities. F.e. results evaluated with the top policy can be artificially boosted and thus they cannot be compared to results evaluated with the ordinal/random or average policy.

## Appendix B. Ruleset Statistics

|            | # C    | # AC1     | # AC2   | # Total   |
|------------|--------|-----------|---------|-----------|
| FB15K-237  | 53,578 | 1,413,806 | 502,593 | 1,969,977 |
| WN18RR     | 7,329  | 38,816    | 25393   | 71,538    |
| YAGO3-10   | 3,699  | 1,955,166 | 333,443 | 2,292,308 |

Table 8: Total number of rules and rule type distributions of rule sets used for the different datasets.

## Appendix C. Runtime comparison AnyBURL - SAFRAN

Table 9 shows a comparison of the runtimes of AnyBURL and SAFRAN, when inferencing filtered predictions of all relevant rules for each prediction task in the test set of a dataset.

The experiment was performed with 8 physical (16 logical) AMD Ryzen 7 PRO 4750U @ 1700 MHz cores and 16 GB of RAM. SAFRAN could drastically improve the runtime of rule application.

| | AnyBURL | SAFRAN | Speed up |
|---|---|---|---|
| FB15K-237 | 2074.0 s | 82.0 s | 25.3 |
| WN18RR | 18.6 s | 3.1 s | 6.0 |
| YAGO3-10 | 3022.0 s | 47.5 s | 63.3 |

Table 9: Comparison of the runtimes

## Appendix D. Threshold Statistics

| | | | $t = 0$ | | $0 < t < 1$ | | $t = 1$ | | $\sim$ | |
|---|---|---|---|---|---|---|---|---|---|---|
| | | | #R | #T | #R | #T | #R | #T | #R | #T |
| FB15K-237 | Grid | Head | 38 | 4890 | 181 | 15393 | 4 | 170 | 14 | 13 |
| | | Tail | 44 | 4479 | 178 | 15944 | 1 | 30 | 14 | 13 |
| FB15K-237 | Random | Head | 21 | 2910 | 202 | 17543 | 0 | 0 | 14 | 13 |
| | | Tail | 34 | 1312 | 189 | 19141 | 0 | 0 | 14 | 13 |
| FB15K-237 | VS | Head | 130 | 14288 | 0 | 0 | 93 | 6165 | 14 | 13 |
| | | Tail | 137 | 14801 | 0 | 0 | 86 | 5652 | 14 | 13 |
| WN18RR | Grid | Head | 2 | 125 | 9 | 3009 | 0 | 0 | 0 | 0 |
| | | Tail | 2 | 27 | 9 | 3107 | 0 | 0 | 0 | 0 |
| WN18RR | Random | Head | 1 | 3 | 10 | 3131 | 0 | 0 | 0 | 0 |
| | | Tail | 1 | 3 | 10 | 3131 | 0 | 0 | 0 | 0 |
| WN18RR | VS | Head | 7 | 2659 | 0 | 0 | 4 | 475 | 0 | 0 |
| | | Tail | 6 | 1368 | 0 | 0 | 5 | 1766 | 0 | 0 |
| YAGO3-10 | Grid | Head | 6 | 1516 | 27 | 3478 | 0 | 0 | 4 | 6 |
| | | Tail | 5 | 38 | 28 | 4956 | 0 | 0 | 4 | 6 |
| YAGO3-10 | Random | Head | 7 | 1835 | 26 | 3159 | 0 | 0 | 4 | 6 |
| | | Tail | 9 | 1597 | 24 | 3397 | 0 | 0 | 4 | 6 |
| YAGO3-10 | VS | Head | 19 | 4510 | 0 | 0 | 14 | 484 | 4 | 6 |
| | | Tail | 18 | 4210 | 0 | 0 | 15 | 784 | 4 | 6 |

Table 10: Number of relations #R and the number of affected test triples #T of each direction (head or tail) for which the Maximum aggregation ($t = 0$), Non-redundant Noisy-OR ($0 < t < 1$) and Noisy-OR ($t = 1$) produced the best results in the validation set. $\sim$ is the number of relations that appear in the training set, but not in the validation set. As the different thresholds could not be evaluated for these relations, the Maximum aggregation is used.

## Appendix E. Clustering Statistics

Table 11 shows statistics of formed clusters for a selection of relations. As can be seen, the average cluster size varies greatly between relations. Extreme examples are the relations *happenedIn* where primarily small clusters were formed and *isAffiliatedTo* where one big cluster and several smaller clusters were formed.

| Relation | #Total Rules | Min | Max | Avg | Std |
|---|---|---|---|---|---|
| created | 212176 | 1 | 1023 | 9.59 | 38.19 |
| isLocatedIn | 164222 | 1 | 868 | 1.55 | 3.08 |
| happenedIn | 18644 | 1 | 47 | 1.61 | 1.75 |
| isAffiliatedTo | 884196 | 1 | 787452 | 16.12 | 3361.89 |

Table 11: Number of total rules and the minimum, maximum, average and standard deviation of sizes of clusters for selected relations of YAGO3-10.

## Appendix F. Experiments with AMIE+

Table 12 shows the results of Non-redundant Noisy-OR applied to rules learned with AMIE+. Rules were learned with default settings, in which AMIE+ only learns cyclic rules. The results can be seen in Table 12. As with rules learned by AnyBURL, Non-redundant Noisy-OR could achieve the biggest performance gain on the FB15k-237 dataset. However, the gains on other datasets are significantly small due to the restriction of the default setting to only cyclic rules. As the rule set only contains only rules of one type (i.e. cyclic rules), only a grid search was performed with the same settings as in Table 7.

| | | MRR | hits@1 | hits@10 |
|---|---|---|---|---|
| FB15K-237 | Maximum | .172 | .121 | .285 |
| | Noisy-OR | .181 | .128 | .291 |
| | NRNO | .186 | .132 | .299 |
| WN18RR | Maximum | .358 | .355 | .362 |
| | Noisy-OR | .358 | .355 | .362 |
| | NRNO | .358 | .356 | .362 |
| YAGO3-10 | Maximum | .338 | .231 | .525 |
| | Noisy-OR | .338 | .231 | .522 |
| | NRNO | .339 | .231 | .526 |

Table 12: Results of Non-redundant Noisy-OR on rules learned by AMIE+.

# Appendix G. Pseudocode of Non-redundant Noisy-OR

---

**Algorithm 1** Non-redundant Noisy-OR: Clustering

---

**Input:** Set of entities $\mathbf{E}$, Set of relations $\mathbf{R}$, Rule set $\mathbf{RS}$, Thresholds $\mathbf{TR}$ of size $|R| \times 6$, Test set $\mathbf{T}$

**for** relation $rel$ in $\mathbf{R}$ **do**                    ▷ Calculation of clusters of redundant rules
    $relevantRules \leftarrow$ rules in $RS$ having relation $rel$ in head;
    **for** $r_i$ in $relevantRules$ **do**
        **for** $r_j$ in $relevantRules$ **do**
            $similarityMatrix[r_i][r_j] \leftarrow jaccard(r_i, r_j)$
        **end for**
    **end for**
    $clusters[rel] \leftarrow \varnothing$
    **for** $r_i$ in $relevantRules$ **do**
        $visited[r_i] \leftarrow 0$
    **end for**
    **for** $r_i$ in $relevantRules$ **do**
        **if** $visited[r_i] = 0$ **then**
            $cluster \leftarrow \varnothing$
            $Q \leftarrow \{r_i\}$
            **while** Q is not empty **do**
                $r_j \leftarrow Q.dequeue()$
                $cluster \leftarrow cluster \cup \{r_j\}$
                $visited[r_j] \leftarrow 1$
                **for** $r_k$ in $relevantRules$ **do**
                    **if** $similarityMatrix[r_j][r_k] > TR[rel][type(r_j, r_k)]$ **then**
                        **if** $visited[r_k] = 0$ **then**
                            $Q.enqueue(r_k)$
                        **end if**
                    **end if**
                **end for**
            **end while**
            $clusters[rel] \leftarrow clusters[rel] \cup \{cluster\}$
        **end if**
    **end for**
**end for**
**for** prediction task $t$ in $\mathbf{T}$ **do**                    ▷ Application of clusters of redundant rules
    **for** $e$ in $E$ **do**
        $nrnoisy[e] \leftarrow 0$
    **end for**
    **for** $cluster$ in $clusters[relation(t)]$ **do**
        **for** $e$ in $E$ **do**
            $maximum[e] \leftarrow 0$
        **end for**
        **for** $rule$ in $cluster$ **do**
            $predictions \leftarrow prediction(rule, t)$
            **for** $e$ in $E$ **do**
                **if** $e \in predictions$ **then**
                    $maximum[e] \leftarrow max(maximum[e], confidence(rule));$
                **end if**
            **end for**
        **end for**
        **for** $e$ in $E$ **do**
            $nrnoisy[e] \leftarrow noisy(nrnoisy[e], maximum[e])$
        **end for**
    **end for**
**end for**

---