# OpenReview forum: "SAFRAN: An interpretable, rule-based link prediction method outperforming embedding models"
_AKBC.ws/2021/Conference — AKBC 2021_

### Official Review · Reviewer_Xz6v · 2021-06-29
**Good paper but can be improved**

**Rating:** 6
**Confidence:** 4

**Review:**

--------

**Summary**

The paper proposes an interpretable rule-based approach for link prediction in knowledge graphs (KGs). The authors identify a common weakness (redundancy) in the following two prior ideas:
* maximum aggregation [Anytime bottom-up rule learning for knowledge graph completion, IJCAI'19]
* noisy-OR aggregation [Fast rule mining in ontological knowledge bases with amie+, VLDB'15].

Experiments on real-world KG datasets demonstrate that a simple combination of the two ideas can overcome redundancy to an extent and compete with state-of-the-art embedding-based / deep-learning-based methods in terms of link prediction performance.

--------

**Quality**

The technical contents of the paper (problem definition, methodology, evaluation) are sound. Claims are well-supported by experiments. However, the authors need to carefully evaluate the limitations of their proposal.

--------

**Clarity**

The illustrations are helpful to understand the main ideas of the approach. The authors have shared the code. A pseudo code would enhance the clarity further and facilitate an expert to reproduce the results of the paper.

--------


**Originality**

The tasks and methods are not new. The main proposal (Non-redundant Noisy-OR) is a simple combination of max aggregation and Noisy-OR which are well-known.  The fact that the simple combination overcomes the common weakness to an extent (Figure 1) is somewhat interesting.

--------

**Significance**

The paper is relevant to the AKBC community. The results seem significant in comparison to rule-learning baselines. However, the paper can be strengthened by
* positioning with explainable KG embeddings,  e.g., (i) Interaction Embeddings for Prediction and Explanation in Knowledge Graphs, WSDM'19, (ii) Faithfully explaining predictions of knowledge embeddings, ENIAC'19 and
* conducting experiments on inductive relation prediction (learned rules can be easily applied to unseen entities at test time), e.g., (i) Communicative Message Passing for Inductive Relation Reasoning, AAAI'21, (ii) Topology-Aware Correlations Between Relations for Inductive Link Prediction in Knowledge Graphs, AAAI'21.

--------

---

> ### Author Response · Authors · 2021-07-29
> **Thank you for your valuable feedback!**
>
> Many thanks for your valuable comments!
>
> > However, the authors need to carefully evaluate the limitations of their proposal.
>
> We have extended the Section “Experiments”.
>
> > A pseudo code would enhance the clarity further and facilitate an expert to reproduce the results of the paper.
>
> We have added a pseudo code to the appendix of the paper.
>
> > positioning with explainable KG embeddings, e.g., (i) Interaction Embeddings for Prediction and Explanation in Knowledge Graphs, WSDM'19, (ii) Faithfully explaining predictions of knowledge embeddings, ENIAC'19 and
>
> We included our positioning with explainable KG embeddings in the Related Work section.
>
> > conducting experiments on inductive relation prediction (learned rules can be easily applied to unseen entities at test time), e.g., (i) Communicative Message Passing for Inductive Relation Reasoning, AAAI'21, (ii) Topology-Aware Correlations Between Relations for Inductive Link Prediction in Knowledge Graphs, AAAI'21.
>
> We tried to apply SAFRAN (Non-redundant Noisy-OR) to the datasets FB15k237_v(1-4) and WN18RR_v(1-4) used in (i) Communicative Message Passing for Inductive Relation Reasoning, AAAI'21, (ii) Topology-Aware Correlations Between Relations for Inductive Link Prediction in Knowledge Graphs, AAAI'21 which were introduced in (iii) Inductive Relation Prediction by Subgraph Reasoning ICML’ 2020. However we ran into some limitations:
>
> * As the set of entities of the training graph and the set of entities of the test graph is disjoint, only rules of type C (cyclic) rules can be used. Rules that contain a constant (regardless if in the head or the body of the rule) cannot be used in these datasets. Such rules are all rules of type AC1 and AC2.
> * Validation sets of these datasets are very small, which leads to overfitting of learned clusters to the validation set. SAFRAN is not able to recognize rule redundancies based on small validation sets. This leads to SAFRAN (NRNO) being not able to exceed the results of the maximum approach. Learning the clusters of rules on the union of validation and test set of the training graph improved the results, sometimes exceeding the Maximum approach for the bigger datasets. To clarify: The test set of the training graph is not to be confused with the test set of the test graph, that is used for evaluation. However, these joint sets are still very small and it is unclear if the authors of the dataset intended the use of the test set of the training graph besides filtering.
>
> As can be seen in the results (following comment), the difference of the Maximum approach and NRNO is less on the bigger datasets (Datasets with a higher version number are bigger) and results on NRNO can exceed the Maximum approach on fb237_v3 and fb237_v4 when clusters of rules are learned on a bigger set of triples (union of validation and test set of the training graph), which is however still too small for NRNO to be able to recognize redundancies.
>
> We also found it difficult to run fair evaluations to compare to results published in the literature, because to calculate hits@10 they compare the score of the positive triple to the scores of 50 randomly sampled negative triples. However, this set of negative triples was never published and using a different set of such triples can heavily impact the evaluation. The results above were evaluated using all possible negative triples in a dataset.

---

> > ### Author Response · Authors · 2021-07-29
> > **Results on inductive link prediction**
> >
> >
> >
> > ```
> >                          MRR   hits1 hits10
> > fb237_v1
> > Max                      0.377 0.317 0.466
> > Grid (NRNO)              0.359 0.300 0.441
> > Grid (NRNO) TestValid*   0.374 0.307 0.463
> > Noisy                    0.355 0.290 0.461
> >
> > fb237_v2
> > Max                      0.483 0.383 0.641
> > Grid (NRNO)              0.447 0.354 0.606
> > Grid (NRNO) TestValid*   0.462 0.356 0.632
> > Noisy                    0.433 0.316 0.645
> >
> > fb237_v3
> > Max                      0.452 0.346 0.623
> > Grid (NRNO)              0.444 0.334 0.619
> > Grid (NRNO) TestValid*   0.454 0.342 0.628
> > Noisy                    0.407 0.284 0.619
> >
> > fb237_v4
> > Max                      0.427 0.322 0.616
> > Grid (NRNO)              0.427 0.318 0.615
> > Grid (NRNO) TestValid*   0.433 0.326 0.620
> > Noisy                    0.380 0.269 0.589
> >
> > WN18RR_v1
> > Max                      0.705 0.652 0.809
> > Grid (NRNO)              0.699 0.644 0.803
> > Grid (NRNO) TestValid*   0.700 0.646 0.803
> > Noisy                    0.676 0.606 0.801
> >
> > WN18RR_v2
> > Max                      0.695 0.633 0.769
> > Grid (NRNO)              0.691 0.627 0.769
> > Grid (NRNO) TestValid*   0.695 0.633 0.769
> > Noisy                    0.689 0.625 0.769
> >
> > WN18RR_v3
> > Max                      0.400 0.352 0.466
> > Grid (NRNO)              0.400 0.352 0.467
> > Grid (NRNO) TestValid*   0.400 0.352 0.468
> > Noisy                    0.400 0.352 0.466
> >
> > WN18RR_v4
> > Max                      0.634 0.585 0.706
> > Grid (NRNO)              0.634 0.584 0.706
> > Grid (NRNO) TestValid*   0.634 0.584 0.706
> > Noisy                    0.626 0.572 0.704
> > ```
> > *Clusters of rules learned on the combination of test and validation set of the training graph. Again: This is not to be confused with the test set of the test graph that is used for evaluation.

---

### Official Review · Reviewer_C4co · 2021-07-21
**Generally a paper with good motivation and considerable performance improvements**

**Rating:** 6
**Confidence:** 3

**Review:**

The paper introduces SAFRAN rule application framework to improve upon the existing AnyBURL interpretable rule-based link prediction approach. SAFRAN helps to solve the redundancies that affect the aggregation step in AnyBURL and introduces Non-redundant Noisy-OR (NRNO) to detects and clusters redundant rules before the aggregation step. In several link prediction benchmarks, SAFRAN achieves SOTA performances for rule-based algorithms and in two out of three datasets even beats multiple embedding-based algorithms with good explainability.

Strengths:
1) The paper is generally well-written and easy to follow, the code is also shared for reproducibility.
2) This paper is well-motivated, in order to reduce the redundant rules which limit the performance of AnyBURL, the author proposes to conduct the rule clustering first based on a redundant estimation metric, then aggregate rules in a cluster using the Maximum approach, and finally aggregate rules in different clusters with Noisy-OR method. The resulting aggregation technique is called Non-redundant Noisy-OR (NRNO), which possesses very good explainability as shown in the Figure 1 example.
3) Experimental results show that SAFRAN achieves state-of-the-art performances over latent embedding-based and interpretable-based approaches on FB15K-237 and WN18RR datasets. While in YAGO3-10,  SAFRAN beats previous interpretable baselines. Overall, the performance improvements look impressive.

Weakness and Questions:
1) Though effective, the NRNO approach proposed in the paper is just a combination of the well-known Maximum and Noisy-OR aggregation methods. Hence, the technical novelty of NRNO is relatively limited.
2) SAFRAN relies on the threshold t_{hd}, which is a list of hyperparameters that may take additional effort to decide with grid search or random search.
3) The paper forgets to compare with HAKE (Zhanqiu Zhang, Jianyu Cai, Yongdong Zhang, and Jie Wang. “Learning hierarchy-aware knowledge graph embeddings for link prediction”, AAAI 2020), which is also a recent competitive baseline. Also, the ComplEx performance numbers in [Rossi et al., 2021] are slightly higher than in Table 6 of the NRNO paper.
4) I didn’t very well understand the meaning of ”VS“ in Table 6. What do the numbers in the VS line indicate? Could you elaborate more on this?
5) There are some minor typos, for example, on Page 4, the fourth last line - “Is is” should be “It is”; on Page 5, the tenth last line - “(i.e)” seems to be redundant, if not it should be “i.e.” instead.

===== After Rebuttal =====

The author's response solves my concerns and I will keep my original rating (6).

---

> ### Author Response · Authors · 2021-07-29
> **Thank you for your valuable feedback!**
>
> Thank you for your time and careful review of our paper.
>
> > Though effective, the NRNO approach proposed in the paper is just a combination of the well-known Maximum and Noisy-OR aggregation methods. Hence, the technical novelty of NRNO is relatively limited.
>
> We agree that the core algorithm of NRNO is rather straightforward. However we want to point out that the application of NRNO is not trivial, as it and symbolic rule based link prediction approaches in general need optimized implementations to achieve reasonable runtimes. As can be seen in Appendix C our implementation accomplishes a significant runtime speed-up in inferring predictions, which is necessary for the computationally expensive hyperparameter search.
>
> > The paper forgets to compare with HAKE (Zhanqiu Zhang, Jianyu Cai, Yongdong Zhang, and Jie Wang. “Learning hierarchy-aware knowledge graph embeddings for link prediction”, AAAI 2020), which is also a recent competitive baseline. Also, the ComplEx performance numbers in [Rossi et al., 2021] are slightly higher than in Table 6 of the NRNO paper.
>
> We have added the results of HAKE, as well as corrected the rounding errors of ComplEx performance numbers.
>
> > I didn’t very well understand the meaning of ”VS“ in Table 6. What do the numbers in the VS line indicate? Could you elaborate more on this?
>
> This experiment is explained in the last paragraph of Chapter 4. With the “VS” experiment we wanted to evaluate how important the optimal thresholds t_{hd} are within Non-redunant Noisy-OR. For this, we evaluated for each relation which aggregation approach (Maximum or Noisy-OR) generates better results on the validation set and apply the best approach for each relation to the test set. Our hypothesis was that this approach should generate better results than simply using the Maximum approach, however Non-redundant Noisy-Or and its combination of both Maximum and Noisy-OR approaches should still produce better results (which is supported by our findings of “VS” in Table 6).
>
> > There are some minor typos, for example, on Page 4, the fourth last line - “Is is” should be “It is”; on Page 5, the tenth last line - “(i.e)” seems to be redundant, if not it should be “i.e.” instead.
>
> Thanks, we have corrected them!

---

### Official Review · Reviewer_nfuz · 2021-07-22
**A focused contribution that shows strong empirical gains**

**Rating:** 7
**Confidence:** 4

**Review:**

This paper studies the problem of link prediction for knowledge graph completion with rule-based methods. Specifically, it builds on a recent strong rule-based method, AnyBURL, for mining rules from a KG, and mainly studies the application of the mined rules. Recognizing the limitations of the Maximum and Noisy-OR rule aggregation strategies used in prior work, this paper proposes SAFRAN, a rule application strategy that clusters rules of the same relation based on their similarity with the hope that redundant rules will be clustered together. The Maximum strategy is then used within each cluster and the Noisy-OR strategy is used to aggregate clusters. Empirical results on FB15K-237, WN18RR, and YAGO3-10 are strong, outperforming embedding-based methods as well as the Maximum and Noisy-OR baselines.

Strengths
- Rule-based link prediction is an important problem with clear advantages in interpret ability
- This paper makes a focused and clear contribution in rule application
- Strong empirical results

Weaknesses
- Sec 2.2, the description of AnyBURL is not very self-contained and one has to resort to the original paper to get a sufficient understanding for the rest of this paper. I understand that we don’t want to spend too much space on repeating prior work, but it could have been described more clearly with the same amount of space. Typos and confusing sentences in this section make it even harder. For example, “ In a rule r uppercase letters represent variables and lowercase letters represent variables.” I believe the second “variables” should be “constants” (after reading the original AnyBURL paper). The next sentence is also very confusing.
- Examples may be taking a bit too much space throughout the paper. It’d be better to use some of the space to have a more in-depth, quantitative analysis of the clustering quality as well as error analysis and case studies. Currently the experiments are a bit thin.

---

> ### Author Response · Authors · 2021-07-29
> **Thank you for your valuable feedback!**
>
> Thank you for your comments on our work, we appreciate your time and effort.
>
> > Sec 2.2, the description of AnyBURL is not very self-contained and one has to resort to the original paper to get a sufficient understanding for the rest of this paper. I understand that we don’t want to spend too much space on repeating prior work, but it could have been described more clearly with the same amount of space. Typos and confusing sentences in this section make it even harder. For example, “ In a rule r uppercase letters represent variables and lowercase letters represent variables.” I believe the second “variables” should be “constants” (after reading the original AnyBURL paper). The next sentence is also very confusing.
>
> Thanks, we have revised this paragraph and made it a bit less abstract.
>
> > Examples may be taking a bit too much space throughout the paper. It’d be better to use some of the space to have a more in-depth, quantitative analysis of the clustering quality as well as error analysis and case studies. Currently the experiments are a bit thin.
>
> This is a good point, we will include this in our future work. We want to add to that, that it is not too trivial to assess the quality of a clustering, as the quality of the clustering would have to be reviewed manually. We have added a remark to this in the conclusion.

---

### Decision · Program_Chairs · 2021-08-17

**Decision:**

Accept

**Comment:**

The reviewers see many merits in the paper, and most of their concerns are addressed by the feedback. We are happy to recommend acceptance. Please make sure to account for all reviewer comments when preparing the camera-ready version.